# Development of a Prediction Model of the Pedestrian Mean Velocity Based on LES of Random Building Arrays

**Sheikh Ahmad Zaki** [1,*] **, Saidatul Sharin Shuhaimi** [1] **, Ahmad Faiz Mohammad** [1] **, Mohamed Sukri Mat Ali** [1] **, Khairur Rijal Jamaludin** [2] **and Mardiana Idayu Ahmad** [3,*]

[1] Malaysia-Japan International Institute of Technology, Universiti Teknologi Malaysia, Kuala Lumpur 54100, Malaysia
[2] Razak Faculty of Technology and Informatics, Universiti Teknologi Malaysia, Kuala Lumpur 54100, Malaysia
[3] Environmental Technology Division, School of Industrial Technology, Universiti Sains Malaysia, Penang 11800, Malaysia
* Correspondence: sheikh.kl@utm.my (S.A.Z.); mardianaidayu@usm.my (M.I.A.)

**Abstract:** Wind speed in urban areas is influenced by the interaction between wind flow and building geometry; at the pedestrian level, the interaction is more complex, particularly with high building density. This study investigated the wind velocity distribution and the mean velocity ratio at the pedestrian level using the large-eddy simulation (LES) database based on random building arrays of several plan area densities, $\lambda_p$. The heights of random buildings are between 0.36 $h$ and 3.76 $h$ where $h = 0.025$ m. Mean streamwise velocity profiles were obtained at the pedestrian level for all arrays and were found to decrease as $\lambda_p$ increased. Wind flow patterns at the pedestrian level were highly influenced by adjacent buildings, especially in denser conditions, $\lambda_p > 0.17$. The pedestrian-level mean velocity was obtained around each building, and the relationship between the local mean velocity ratio, $V_{p(t)}$ and the local frontal area density, $\lambda_{f(t)}$ was analyzed. Subsequently, a prediction model was formulated based on the building's aspect ratio, $\alpha_p$; the correlation for high-rise buildings with 2.64 $h \leq \alpha_p \leq 3.76$ $h$ was high at 0.8, while a lower correlation was obtained for lower buildings due to random positioning and surrounding geometric effects. Therefore, the impact of high-rise buildings on pedestrian wind velocity can be estimated more accurately using the formulated model.

**Keywords:** pedestrian wind; random building array; large-eddy simulation; plan area density; prediction model

## 1. Introduction

Natural wind in urban environments can improve the quality of air as it facilitates the removal of pollutants and greenhouse gasses. A well-ventilated urban environment can also help alleviate pedestrian discomfort which is linked to the urban heat island (UHI) phenomenon, wherein the temperature in the urban area is higher than in the surrounding area [1]. This is a common phenomenon occurring in urban areas around the world [2,3]; urban areas are generally made up of concrete buildings and asphalt roads that absorb heat from sunlight radiation, thereby increasing the air temperature. Moreover, this is worsened due to minimal ventilation in the urban area since the flow of natural wind is inhibited by the presence of buildings and other impeding structures [2].

The geometry of an urban area can be described by various geometrical factors, including building height variation, building arrangement, and density of the area. Hence, various wind flow studies conceptualize these structures into "idealized urban models", where all forms of parameters are incorporated to assess their influence on urban wind flows using either wind tunnel experiments [4,5] or computational fluid dynamics (CFD) simulations [6–9]. Several geometrical parameters of an urban area are investigated with relation to changes in wind flows. The plan area density, $\lambda_p$, which is defined as the ratio of the total planar area of buildings over the total floor area, has been associated with the

changes in the wind flow velocity; the wind flow velocity tends to decrease with an increase in $\lambda_p$ due to surface friction [4,9]. The vertical profile of mean wind velocity over rough surfaces typically shows a decreasing pattern when $\lambda_p$ increases [4–6].

In addition, the variation in building heights which is parameterized by the aspect ratio, $\alpha_p$, also affects the wind flow behaviour or pattern in urban areas [7,10]. There are generally three types of wind flow regimes formed between two adjacent buildings, as introduced by Oke [11]; isolated roughness flow, wake interference flow, and skimming flow. The different flow regimes have been extensively studied in relation to wind-induced ventilation, pedestrian-level wind, and pollutant dispersion, amongst others. A study by Hang et al. [7] used uniform and non-uniform height arrays with $\lambda_p = 0.25$ and 0.4 in WTE and CFD simulations and demonstrated the occurrences of different flow regimes. They concluded that the ventilation rate decreases due to the skimming flow behaviour in denser conditions.

Furthermore, the configuration of buildings in an urban area is influential on the wind flow behaviour. For example, it was discovered that a staggered array imposes less obstruction to wind flows than a square array of the same $\lambda_p$. For example, Cheung and Liu [12] found that in a staggered array, an increase in the ventilation rate was observed, relative to the square array of the same $\lambda_p$. Furthermore, Hagishima et al. [5] and Kanda et al. [9] found that the drag coefficient is highly related to $\lambda_p$ for staggered arrays rather than with square arrays.

With the available findings of how building arrangements can affect wind flows, more work has been devoted to assessing wind flow behaviour at a pedestrian level. Razak et al. [10] used both uniform and non-uniform building heights in their arrays to investigate the pedestrian wind velocity in relation to the frontal area density, $\lambda_f$, which is the ratio of the frontal area to the plan are of the building; their findings demonstrated the suitability of using $\lambda_f$ in relation to the mean velocity at the pedestrian level. This is further supported by the findings of a residential study in Tianjin where the pedestrian level wind was found to be correlated to $\lambda_f$ as the area consists of buildings of various heights [13]. Moreover, Xie et al. [8] demonstrated that velocity increases more profoundly around taller buildings due to the channelling effect. A study of high-rise buildings demonstrated that the decrease or increase in pedestrian-level wind speed is mainly altered by the surrounding buildings [14].

Several studies proposed prediction models using urban geometric parameters to estimate the wind velocity at the pedestrian level. Hereby, the prediction models are consistently evaluated using the mean velocity ratio taken at numerous points around the respective modelled arrays. Formerly, a study by Kubota et al. [15] conducted wind tunnel tests on several residential areas in Japan where the data of the wind velocity were obtained for all cases. Then, the results of the mean velocity ratio were plotted by normalizing the wind speed taken at pedestrian height ($V_p$) with those at the same height but with no models in ($V_{no}$). Therefore, $\lambda_f$ was found to strongly influence the mean velocity ratio in the majority of the cases. This was also obtained in the study by Razak et al. [10], in which several idealized cases of arrays exhibiting a variety of geometrical parameters using cubical and rectangular blocks with different plan area densities were used. Here, the wind speed at the pedestrian level ($V_p$) is normalized with those taken at a height twice the highest building height ($V_{2hmax}$) where the data displays the same condition with the case without building models. The trend was found to be exponential and able to fit a wide range of plan area densities and various building heights.

The database of Razak et al. [10] was utilized by Ikegaya et al. [16] to be further evaluated, as it is believed the prediction model of the mean velocity ratio would be more conclusive if it accounts for the $\lambda_p$ ranging from zero to one. Accordingly, a new geometrical parameter ($\varsigma$) was derived empirically to represent the function of $\lambda_p$ and $\alpha_p$, which leads to improvement in the prediction of the mean velocity ratio. A prediction model was constructed based on the parameter and is targeted to fit the $\lambda_p$ range sufficiently but it is currently only applicable to arrays consisting of uniform-height buildings only. The overview of the studies on the prediction model of the mean velocity ratio reviewed is summarized in Table 1.

**Table 1.** Overview of studies on the prediction model of mean velocity ratio at pedestrian level.

| Author | Prediction Model | Findings | Layout | $\lambda_p$ | $\lambda_f$ | $\lambda_{f(t)}$ | $\alpha_p$ | UH | N-UH | Method |
|---|---|---|---|---|---|---|---|---|---|---|
| Kubota et al. [15] | $\frac{V_p}{V_{no}} = -A\lambda_f + B$ | For : $0.2 < \lambda_f < 0.7$<br>A = 0.4 and B = 0.55 | Real urban case | ✗ | ✓ | ✗ | ✓ | ✗ | ✓ | WTE |
| Yoshie et al. [17] | $\frac{V_p}{V_{BL}} = -A\lambda_p + B$ | Wind speed at boundary layer ($V_{BL}$)<br>A = 0.2 and B = 0.273 | Real urban case | ✓ | ✗ | ✗ | ✓ | ✗ | ✓ | WTE |
| Tahbaz et al. [18] | $\frac{\overline{V_z}}{\overline{V_{z_{10}}}} = \left[\frac{Z}{Z_{10}}\right]^\alpha$ | Mean wind speed at height z, ($V_z$), Height 10 m ($Z_{10}$),<br>Surface roughness ($\alpha$) | Isolated building | ✗ | ✗ | ✗ | ✓ | ✗ | ✗ | Graphical method |
| Razak et al. [10] | $\frac{V_p}{V_{2hmax}} = A\lambda_f^{-B}$ | For : $0.15 < \lambda_f < 0.67$<br>A = 0.025 and B = 0.8 | Staggered array | ✓ | ✓ | ✗ | ✓ | ✓ | ✓ | CFD |
| Ikeda et al. [19] | $\frac{V_p}{V_{2hmax}} = A\alpha_p^{-b}\lambda_p^{-c}$ | For regions: Front, behind and sides of a building | Staggered array | ✓ | ✗ | ✗ | ✓ | ✓ | ✗ | Database evaluation, mathematical derivations |
| Yuan et al. [20] | $VR = -3.9\lambda_{f_{point}} + 0.41$ | Wind velocity ratio (VR), Point specific $\lambda_f (\lambda_{f\_point})$ | Real urban case | ✓ | ✓ | ✓ | ✗ | ✗ | ✓ | Modelling-mapping |
| Ikegaya et al. [16] | $\frac{V_p}{V_{2hmax}} = A(1-\varsigma)^B$ | For : $\lambda_f < 1$;<br>$\varsigma(\lambda_p, \alpha_p, z) = 1 - (1 - \lambda_p)^{\alpha_p^a}$ | Staggered array | ✓ | ✓ | ✗ | ✓ | ✓ | ✗ | Database evaluation, mathematical derivations |
| Ikegaya et al. [21] | $\varphi_F^n = p\sigma_\varphi + \overline{\varphi}$ | Pedestrian exceeding wind speed ($\varphi_f$), Peak factor ($p$),<br>Std dev of wind speed ($\sigma_\varphi$) | Isolated building | ✗ | ✗ | ✗ | ✓ | ✗ | ✗ | Database evaluation, mathematical derivations |
| Weerasuriya et al. [22] | Gaussian-process emulator | Lift-up building, for indoor & outdoor wind | Isolated building | ✗ | ✗ | ✗ | ✓ | ✗ | ✗ | CFD |

Key—$\lambda_p$: Plan area density, $\lambda_f$: Frontal area density, $\lambda_{f(t)}$: Local frontal area density, $\alpha_p$: Building's aspect ratio, UH: Uniform building height, N-UH: Non-uniform building height, WTE: Wind tunnel experiment, CFD: Computational Fluid Dynamics, '✗': Not related or discussed in the listed study, '✓': Related and discussed in the listed study.

Although various prediction models have been proposed to meet various urban conditions, there has not been a credible model that can predict the mean velocity ratio in an urban area exclusively around a target building. Existing prediction models of the mean velocity ratio are generally applied on the spatial basis of an urban array but not exclusively around individual buildings which should be highlighted considering the configuration of urban arrays are typically random.

In addition, numerical simulations were performed by Mohammad et al. [23] on random staggered arrays, which are the idealized urban form. The usage of the simulation database of Mohammad et al. [23] has been put into interest for the evaluation of the pedestrian wind environment. The use of the random staggered arrays highlights the variation in building heights on a larger scale. This brings us to the aims of this study, which are: (1) to examine the pedestrian wind velocity distribution in the random staggered arrays and (2) to propose a prediction model of the mean velocity ratio locally at the pedestrian level using the building's frontal area density. It is important to note the wind velocity at the pedestrian level within an urban array tends to vary with location, as it is bounded by buildings of different heights. The rest of this paper is structured as follows; Section 2 describes the main configurations and settings of the random staggered array; Section 3 discusses the effect of the random staggered array, including the proposed prediction model to evaluate the local mean velocity ratio; lastly, Section 4 summarizes the outcome of the study from the results obtained.

## 2. Methodology

This study uses the large-eddy simulation (LES) results obtained by Mohammad et al. [23] on the random staggered building arrays. The database of their simulation results contained wind velocity data of six random staggered building arrays. In the work of Mohammad et al. [23], the LES results of mean wind velocity and pressure were validated and compared for accuracy with respect to experimental and numerical results. The accuracy of their LES results was thoroughly demonstrated, and this makes the work of Mohammad et al. [23] feasible and suitable to be extended and applied in this study.

### 2.1. Building Type and Geometry

Referring to Mohammad et al. [23], all six arrays consisted of nine types of buildings, denoted as *B1* to *B9*, where each has a different height, as provided in Table 2; *h* is the height unit equivalent to 0.025 m and the aspect ratio of a building, $\alpha_p$ is defined as the ratio of the building's frontal area to its planar area. The plan area of all buildings is the same, each having a square base ($h \times h$). The inclusion of nine building types is feasible for this study to universally represent real urban conditions for the analysis of pedestrian-level wind environment.

**Table 2.** Height of building blocks where *h* = 0.025 m.

| Building | Height | Aspect Ratio, $\alpha_p$ | Remark |
|:---:|:---:|:---:|:---:|
| B1 | 0.36 *h* | 0.36 | Low-rise |
| B2 | 0.84 *h* | 0.84 | |
| B3 | 1.32 *h* | 1.32 | Medium-rise |
| B4 | 1.50 *h* | 1.50 | |
| B5 | 2.00 *h* | 2.00 | |
| B6 | 2.64 *h* | 2.64 | High-rise |
| B7 | 3.00 *h* | 3.00 | |
| B8 | 3.32 *h* | 3.32 | |
| B9 | 3.76 *h* | 3.76 | |

Figure 1 shows the schematic of a random staggered array, which has 25 buildings in total. In relation to real urban buildings, low-rise buildings are commonly classified as those with less than five storeys, medium-rise buildings are in the range of five to ten storeys,

whereas high-rise buildings have generally more than ten storeys [24]. The buildings are arranged in a staggered manner, whereby the tallest building, *B9* is positioned at the centre of the array. Notably, low-rise and medium-rise buildings are positioned to surround the centre, whereas the rest of the tall-rise buildings are placed in close proximity to the tallest building, replicating a typical urban condition [25].

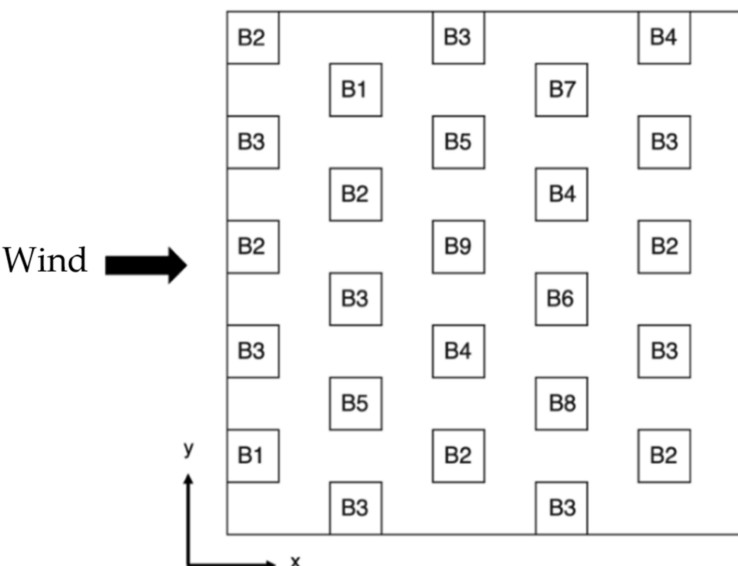

**Figure 1.** Plan view of the configuration of the random staggered array with wind flow in the x direction for $\lambda_p$ = 0.25 [24].

*2.2. Plan Area Density*

The six random staggered arrays were differed by the plan area density, $\lambda_p$, as shown in Table 3. The building densities of the arrays ranged from a sparse condition ($\lambda_p$ = 0.04) to a dense condition ($\lambda_p$ = 0.39). The arrangement of buildings remains fixed for all arrays. For clarity, the name of each simulation case is standardized in the same format; as an example, for "R4A", the letter "R" denotes "random staggered array", which is followed by the value of $\lambda_p$ (i.e., 0.04), and "A" represents the wind direction, which is along the x-axis. Because all buildings have the same plan area, the $\lambda_p$ of each random staggered array is the same as a building's local plan area density of that particular array; this is a similar geometric configuration set up for non-uniform building arrays in previous studies [4,10].

**Table 3.** List of simulation cases and the computational domain size [23].

| Case | $\lambda_p$ | Computational Domain Size ($L_x \times L_y \times L_z$) |
|---|---|---|
| R4A | 0.04 | $24\,h \times 24\,h \times 15\,h$ |
| R8A | 0.08 | $17.5\,h \times 17.5\,h \times 15\,h$ |
| R17A | 0.17 | $12\,h \times 12\,h \times 15\,h$ |
| R25A | 0.25 | $10\,h \times 10\,h \times 15\,h$ |
| R31A | 0.31 | $9\,h \times 9\,h \times 15\,h$ |
| R39A | 0.39 | $8\,h \times 8\,h \times 15\,h$ |

Figure 2 shows the schematic of the computational domain used for all six arrays, adapted from Mohammad et al. [23]. The vertical height of the computational domain, $L_z$ is 4 $h_c$ where $h_c$ is the height of the tallest building, i.e., *B9*(3.76 *h*), while the horizontal domain size (streamwise length, $L_x$ and spanwise length, $L_y$) is adjusted to each $\lambda_p$, as shown in Table 3. The boundary conditions of the computational domain are as follows: a cyclic condition is applied in the streamwise direction to simulate a continuous wind flow; a symmetry condition in the lateral direction; a free-slip condition on the top boundary; the no-slip condition on wall surfaces of the buildings and the domain's floor. The minimum

near-wall grid size is $h/14$. This was considered adequate for estimating the average wind velocity inside the canopy.

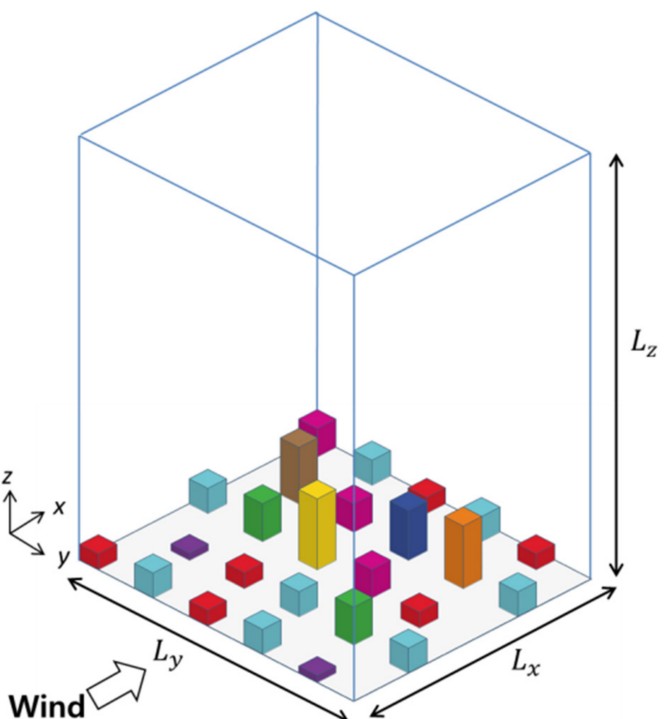

**Figure 2.** Schematic diagram of the computational domain for $\lambda_p$ = 0.25. [23]. Different colors indicate different building types: *B1* (purple), *B2* (red), *B3* (light blue), *B4* (magenta), *B5* (green), *B6* (dark blue), *B7* (brown), *B8* (orange), and *B9* (yellow).

### 2.3. Frontal Area Density

The frontal area density, $\lambda_f$, is an important parameter characterizing the building density of vertically random building arrays. It is defined as the ratio between the total frontal area of all buildings, $A_F$, to the total ground surface area of an array, $A_S$ [10,16]:

$$\lambda_f = A_F/A_S \tag{1}$$

Since a random staggered building array contains 25 buildings of nine varying heights, a more specific geometric parameter characterizing each building is needed for analysis. Therefore, the local frontal area density, $\lambda_{f(t)}$, is defined as the product of the aspect ratio of the respective building, $\alpha_p$, with the value of $\lambda_p$ of an array, given by Equation (2):

$$\lambda_{f(t)} = \alpha_p \times \lambda_p \tag{2}$$

The local mean velocity ratio taken at the pedestrian level around each building, $V_{p(t)}$, is also introduced; this is needed to establish a correlation with $\lambda_{f(t)}$, which is elaborated in Section 3. Figure 3 illustrates an example of the mean velocity ratio taken around *B4* (yellow) in a random staggered array. $V_{p(t)}$ is obtained by averaging the mean streamwise velocity at the pedestrian level, $V_p$, at locations marked with "x" within the shaded region.

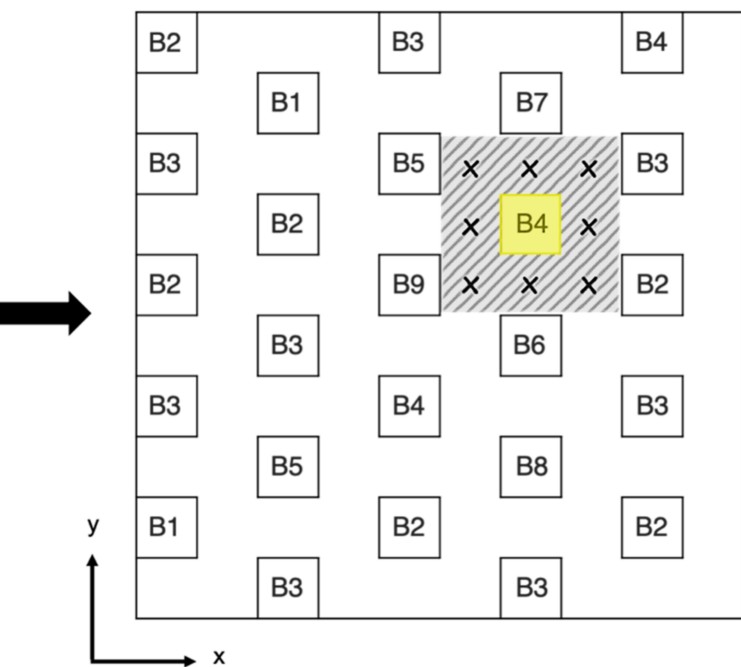

**Figure 3.** Example of how the local mean velocity ratio is determined around *B4* for R25A ($\lambda_p = 0.25$). The arrow indicates the wind flow direction, '×' indicates the location where the mean pedestrian-level wind velocity around a building is extracted, and building labels '*B1*' to '*B9*' represent different building types: *B1* ($\alpha_p = 0.36$), *B2* ($\alpha_p = 0.84$), *B3* ($\alpha_p = 1.32$), *B4* ($\alpha_p = 1.5$), *B5* ($\alpha_p = 2.0$), *B6* ($\alpha_p = 2.64$), *B7* ($\alpha_p = 3.0$), *B8* ($\alpha_p = 3.32$), and *B9* ($\alpha_p = 3.76$).

## 3. Results

### 3.1. Mean Velocity Field at Pedestrian Level

The mean wind velocity is analysed at the pedestrian level, which is a region within the height of less than 2 m on the real scale [15]. Since the random staggered arrays were used, the pedestrian level at 1.5 m was chosen following Razak et al. [10], which also adopted LES and staggered building arrays in their work. From the LES database, the mean wind velocity was extracted as point data at the selected pedestrian height throughout the computational domain for all six cases.

Figure 4 illustrates the mean velocity distribution on the horizontal plane at the pedestrian level for all cases ($0.04 \leq \lambda_p \leq 0.39$). By overall comparison, it can be seen that the velocity distribution is affected by the presence of buildings and the mean velocity decreases as $\lambda_p$ increases. This can be explained in-depth from individual figures. In Figure 4a where $\lambda_p = 0.04$, it can be seen that the mean velocity can reach 4 m/s or more in the streamwise direction and remains high throughout the domain with the exception of local downstream regions of the buildings where reverse flows form. The mean velocity distribution around all buildings in the array is relatively similar, and this is also observed in Figure 4b for $\lambda_p = 0.08$. For these two cases, wherein the buildings are sparsely positioned, the relatively large size of the street canyon imposes minimal obstruction to the flow and allows a continuous stream throughout the domain. In addition, a slight increase in velocity is observed on the lateral sides of taller buildings (*B5* to *B9*), for example in a region between *B4* and *B9*, due to the channelling effect.

In Figure 4c,d, the velocity is relatively lower than in sparse conditions, e.g., $\lambda_p = 0.08$. This is due to the increased interaction between the wind flow and the buildings; the separation distance between adjacent buildings is decreased, thereby limiting the movement of the wind flow. Moreover, the mean velocity in vast regions of $\lambda_p = 0.31$ and $\lambda_p = 0.39$ shown in Figure 4e,f, respectively, is less than 1 m/s, which is relatively low due to the high density of the built-up area. When $\lambda_p$ increases to 0.39, the separation distance between the buildings is reduced, hence the flow tends to skim over the buildings. For the random

staggered arrays, the formation of flow becomes more complicated since the flow around each building is altered by adjacent buildings. Meanwhile, the mean velocity between the buildings in the fourth row is the highest in the arrays as a result of contraction flow [26].

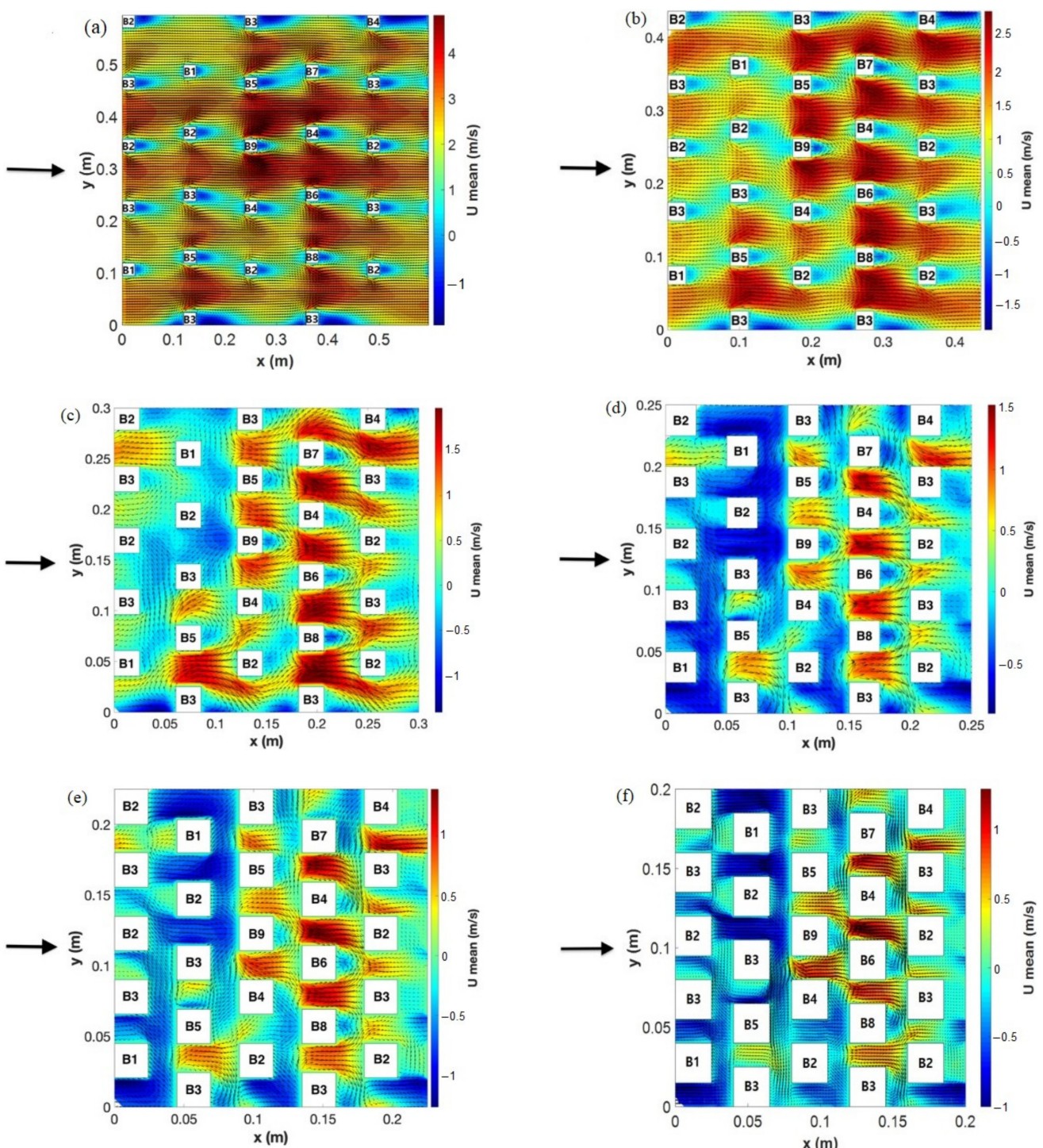

**Figure 4.** Mean wind velocity distribution on the horizontal plane at the pedestrian level for (**a**) R4A ($\lambda_p = 0.04$), (**b**) R8A ($\lambda_p = 0.08$), (**c**) R17A ($\lambda_p = 0.17$), (**d**) R25A ($\lambda_p = 0.25$), (**e**) R31A ($\lambda_p = 0.31$), and (**f**) R39A ($\lambda_p = 0.39$). The arrow indicates the wind direction and the labels for building types (i.e., *B1* to *B9*) are displayed for all buildings.

In Figure 5, the mean velocity distribution is shown for the densest array, i.e., R39A ($\lambda_p = 0.39$), at a vertical cross-section along the *y*-axis (*y* = 0.12 m). The figure shows the

occurrence of a skimming flow in front of *B9* creating a large vortex behind the building approximately at 0.06 m. Due to the staggered building arrangement, the vortex behind *B9* does not attach to the flow around the downstream building, *B2* due to a large difference in building height. In addition, the observed skimming flow occurs because the wind flow skims over *B2* and is obstructed by *B9*, creating a downdraft. The figure shows that the presence of a high-rise building can significantly alter the local flow near its neighbouring buildings.

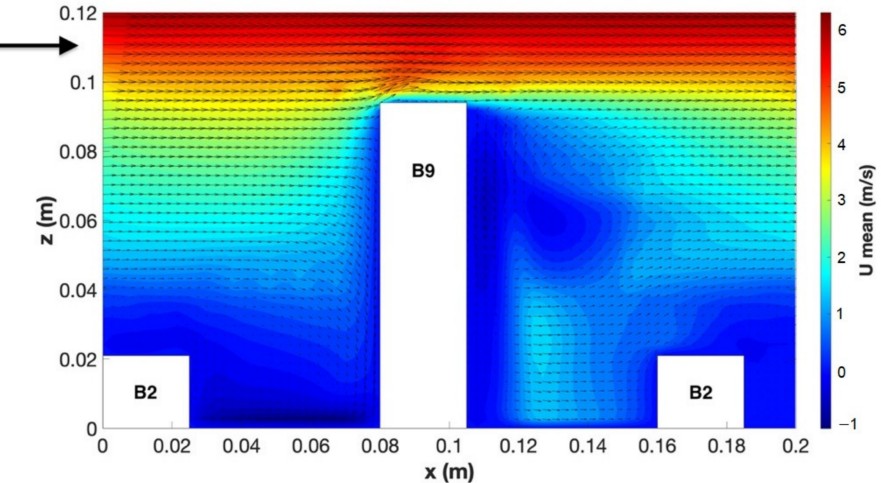

**Figure 5.** Mean wind velocity distribution along the vertical cross-section at *y* = 0.12 m for R39A ($\lambda_p$ = 0.39). The arrow indicates the wind direction and the targeted buildings shown are *B2* ($\alpha_p$ = 0.84) and *B9* ($\alpha_p$ = 3.76).

*3.2. Influence of Plan Area Density and Frontal Area Density on Mean Velocity Ratio*

The mean wind velocity ratio, denoted by $V_p/V_{2hmax}$, is the spatially averaged wind speed taken at the pedestrian level ($V_p$), normalized by the wind speed at twice the maximum building height ($V_{2hmax}$). Figure 6 shows the results of $V_p/V_{2hmax}$ plotted against $\lambda_f$.

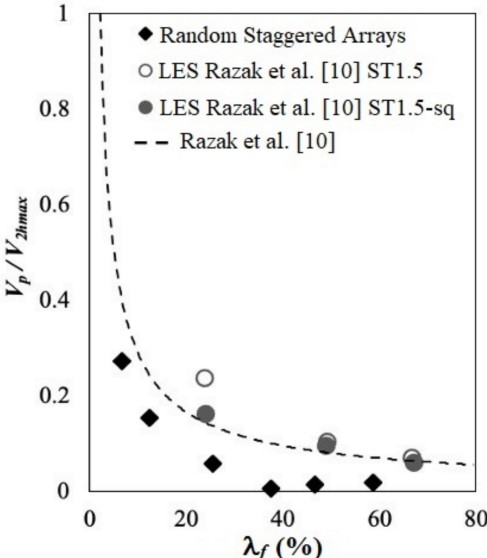

**Figure 6.** Relationship between $V_p/V_{2hmax}$ and the frontal area density, $\lambda_f$ for the random staggered arrays from the LES database [23], compared with the results of Razak et al. [10].

In Figure 6, the predicted results obtained by the equation model proposed by Razak et al. [10] are included. It is apparent that the mean velocity ratio obtained from the

random staggered array of the current study (represented with black diamonds) has a large discrepancy with the model of Razak et al. [10]. This highlights the incompatibility of the model equation proposed by Razak et al. [10] with the random staggered arrays. Although the random array used by Razak et al. [10], i.e., ST1.5-sq has similar characteristics with those of the random staggered arrays used in this study, i.e., it is comprised of buildings arranged in a staggered layout and has the average building height ($h_{ave}$) of 1.5 $h$. Moreover, ST1.5-sq is made up of buildings with only two different heights. The developed equation model is therefore not suitable for the random staggered arrays that have more than two building height variations. Nevertheless, it can be seen that the use of $\lambda_f$ as the correlation parameter yields a similar decreasing pattern of the mean velocity ratio with the other random arrays shown.

*3.3. Mean Velocity Ratio around Various Buildings*

Within the random staggered array, a total of nine building heights correspond to nine different aspect ratios, $\alpha_p$. $\lambda_f$ is obtained by averaging the $\alpha_p$ of all buildings in an array; it is the main parameter in the equation model of Razak et al. [10]. It is expected that the reliability of the prediction model is highly dependent on the number of buildings in the array, thus affecting the value of $\lambda_f$. The mean flow velocity tends to vary in local regions around buildings of varying heights. Due to this, the local frontal area density, $\lambda_{f(t)}$ introduced in Section 2.3 is used for correlation with the mean velocity ratio around individual buildings.

Additionally, for a more accurate correlation, the local mean velocity is used and normalized with $V_{2hmax}$. This is carried out for all twenty-five buildings in the array and repeated for each case of $\lambda_p$. Figure 7a shows the relationship between $V_{p(t)}$ and $\lambda_{f(t)}$ for all buildings in the random staggered arrays. The prediction models proposed in previous studies by Razak et al. [10] and Kubota et al. [15] are also plotted together for comparison. The coefficient of determination, $R^2$ obtained is 0.45, suggesting a low correlation when all the buildings are considered. It is clear that $V_{p(t)}$ decreases when $\lambda_{f(t)}$ is increased, and it can be seen that $V_{p(t)}$ is influenced by the $\alpha_{p(t)}$ of a building under the same $\lambda_p$. This is explained as the equation obtained by Razak et al. [10], which shows a reasonable trend with the current LES results, but the discrepancy is seen, especially in smaller values of $\lambda_{f(t)}$ for each condition of $\lambda_p$.

The low regression obtained in the plot is partly due to the large range of building heights of the random staggered array. Therefore, to improve the regression of the data, the results are classified according to the height range, namely, low-rise buildings (0.36 $h$ to 0.84 $h$), medium-rise buildings (1.32 $h$ to 2.00 $h$) and tall-rise buildings (2.64 $h$ to 3.76 $h$); the results of all the building categories are plotted in Figure 7b–d, respectively.

The correlation is improved for these categories, whereby the highest regression obtained through the power law equation is around high-rise buildings ($R^2$ = 0.80). Although the trend shown is typically similar for all three categories, the dispersion of $V_{p(t)}$ data around higher buildings in the random staggered array is more accounted for in all cases of $\lambda_p$. In the case of sparse conditions ($\lambda_p < 0.25$), the $V_{p(t)}$ around tall-rise buildings is relatively large due to the isolated roughness flow; $V_{p(t)}$ exponentially decreases due to the change in the flow regime (from the isolated roughness flow to the wake interference and skimming flows) when $\lambda_p$ is increased.

Meanwhile, for the low-rise and medium-rise buildings, the correlation is considered moderate, where $R^2$ = 0.54 and $R^2$ = 0.60, respectively. The dispersion of data is less dependent on $\lambda_{f(t)}$ due to obstruction and change in flow induced by the taller buildings, which causes either wake interference flow or skimming flow to happen around them. The low-rise and medium-rise buildings that are located near high-rise buildings are more likely to experience lower wind speeds [27]. This also applies strongly to buildings on the front rows of the array and the effect becomes more significant when $\lambda_p$ is increased. This is due to the increase in drag at high $\lambda_p$ [28,29]; the buildings are more closely spaced and

the interaction between the wind flow and the buildings results in lower wind speed at the pedestrian level.

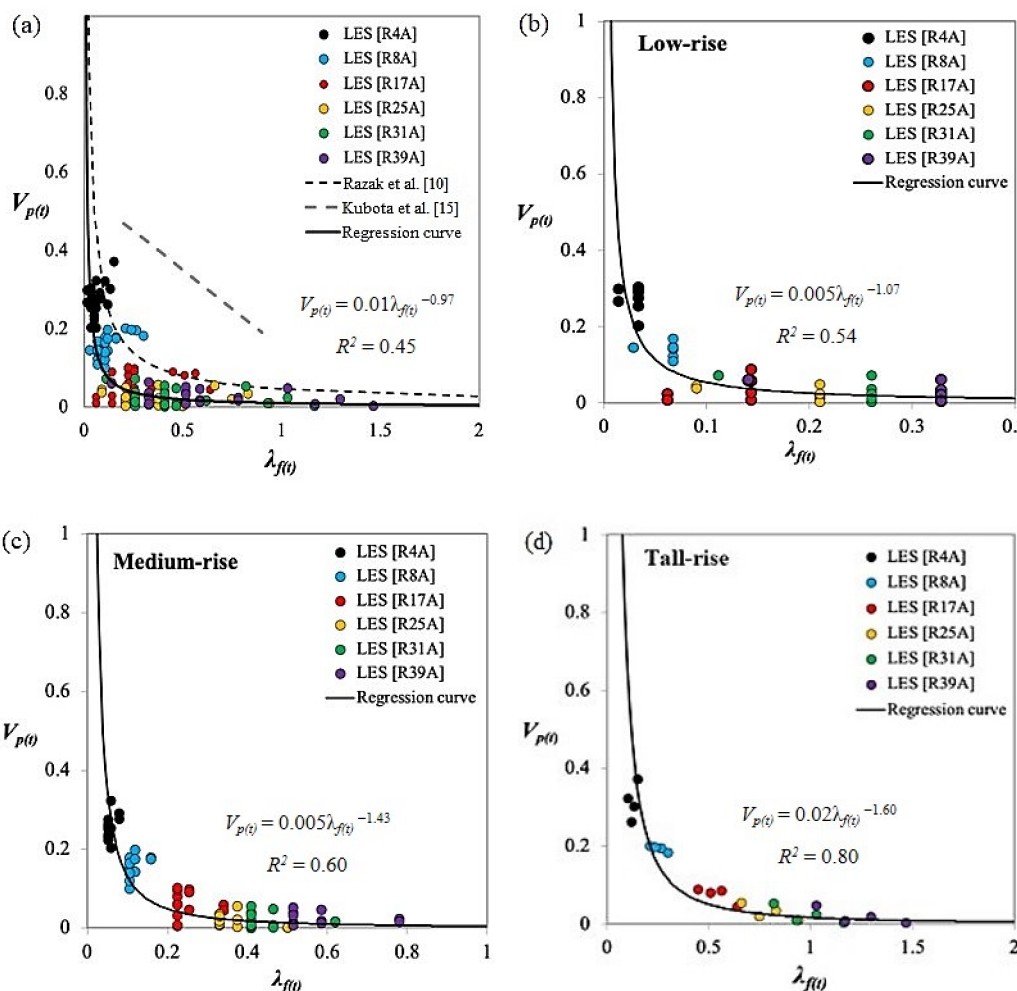

**Figure 7.** (**a**) Relationship between $V_{p(t)}$ and $\lambda_{f(t)}$ for each building of all random staggered arrays, (**b**) low-rise, (**c**) medium-rise, and (**d**) high-rise buildings.

### 3.4. Prediction Model for Local Mean Velocity Ratio

Overall, the parameterization of the local frontal area density, $\lambda_{f(t)}$ and the local mean velocity ratio, $V_{p(t)}$ contributes to the most ideal model to estimate the pedestrian wind velocity in the random staggered array. The relationship between $V_{p(t)}$ and $\lambda_{f(t)}$ can be expressed in a power law equation based on a building's aspect ratio in Equation (3), as shown below:

$$V_{p(t)} = \frac{V_p}{V_{2hmax}} = A_{(t)}\lambda_{f(t)}^{-(B_{(t)})} \tag{3}$$

where, $V_{p(t)}$ is the local mean velocity ratio of a respective building of interest, $V_p$ is the streamwise velocity around a targeted building, and $V_{2hmax}$ is the streamwise velocity at two times the maximum building height of the array. The empirical constants, $A_{(t)}$ and $-B_{(t)}$, are based on the building's aspect ratio and varied for different building categories: (a) the low-rise buildings ($0.36h < \alpha_p < 0.84h$), $A_{(t)} = 4.50 \times 10^{-3}$ and $-B_{(t)} = 1.07$; (b) the medium-rise buildings ($1.32h < \alpha_p < 2.00h$), $A_{(t)} = 4.90 \times 10^{-3}$ and $-B_{(t)} = 1.07$; and (c) the tall-rise buildings ($2.64h < \alpha_p < 3.76h$), $A_{(t)} = 0.02$ and $-B_{(t)} = 1.60$.

This model is suitable to estimate the pedestrian mean velocity ratio on buildings in an urban array exhibiting non-uniform building heights. In relation to the real urban conditions, it is deemed appropriate for the prediction model to be applied for $0.04 \leq \lambda_p \leq 0.4$

since the results of this study satisfy these conditions. The developed prediction model can be potentially used in typical urban areas that are generally within $0.25 \leq \lambda_p \leq 0.5$ [7]. The prediction model is still considered reasonable to be used in conditions with $\lambda_p < 0.25$ because the interference effects caused by the surrounding buildings are not significant in sparse conditions [24,28].

The use of the local frontal area density of an individual building in developing the prediction model implies that interference effects caused by surrounding buildings are considered. This is important because the pedestrian-level wind is varied by location within an urban array, especially in the upwind, downwind and lateral sides of a building. In the case of a random staggered array, greater interference effects are caused by adjacent buildings, especially when $\lambda_p$ is high. Hence, by calculating the mean velocity ratio of individual buildings, it can provide more reliable and more accurate estimations.

## 4. Conclusions

This study adopted the LES database of the previous study based on the random staggered building arrays to analyse the pedestrian wind environment and develop a prediction model. Firstly, it was found that the mean wind velocity obtained at the pedestrian level was affected by the plan area density; the visualizations of the mean velocity distribution indicated a gradual decrease in the pedestrian-level mean velocity among buildings with the increase in the plan area density; for example, the maximum mean velocity which was around 4 m/s in the most sparse array, i.e., R4A ($\lambda_p = 0.04$), was reduced to 1 m/s in the most dense array, i.e., R39A ($\lambda_p = 0.39$). In addition, buildings with a higher aspect ratio (e.g., $\alpha_p = 3.76$) significantly obstruct the wind flow, causing changes in the wind flow direction and velocity in the respective windward and leeward regions; this impact is especially significant in denser urban conditions ($\lambda_p \geq 0.25$). This helps to determine the importance of a building's position in an array. The results showed that the wind conditions surrounding low-rise buildings are greatly affected due to the flow interference caused by taller adjacent buildings.

Next, the mean velocity ratio, $V_{p(t)}$, at the pedestrian level was used for correlation with the frontal area density. The comparison among the random staggered arrays and non-uniform arrays from the previous study showed a similar tendency for mean velocity to decrease with the increase in the frontal area density; however, the prediction model developed in the previous study was shown to be less accurate and not suitable for the random staggered arrays. Therefore, a new relationship, which is between $V_{p(t)}$ and the local frontal area density, $\lambda_{f(t)}$, was formulated. It was found that the prediction of the mean velocity ratio which was evaluated locally for individual buildings provided more accurate results by considering the individual buildings' effects on the pedestrian wind speed. Furthermore, the relationship of the mean velocity ratio around tall-rise buildings ($2.64 \leq \alpha_p \leq 3.76$) resulted in the highest correlation with $R^2 = 0.80$. Nevertheless, the predictability of the formulated model is relatively low for medium-rise and low-rise buildings. The lower buildings were shown to be more affected by the presence of taller buildings, particularly in denser arrays ($\lambda_p \geq 0.25$). However, the predictability of the formulated model can perhaps be improved in future work by using more simulation data supported by wind tunnel experimental data.

In summary, a prediction model for the pedestrian-level mean velocity ratio was formulated with the empirical constants classified based on the aspect ratios of the buildings. The model can be adequately utilized in predicting the mean velocity ratio at the pedestrian level around buildings in various urban arrays, particularly those with non-uniform building heights.

**Author Contributions:** Data curation, S.S.S.; Formal analysis, S.S.S.; Funding acquisition, S.A.Z., M.I.A., M.S.M.A. and K.R.J.; Investigation, S.S.S.; Methodology, S.S.S.; Project administration, S.A.Z.; Supervision, S.A.Z. and A.F.M.; Validation, S.S.S.; Writing—original draft, S.S.S.; Writing—review and editing, A.F.M. and S.A.Z. All authors have read and agreed to the published version of the manuscript.

**Funding:** This research was financially supported by the UTM Fundamental Research Grant (21H16), Industry-International Incentive Grant from Universiti Teknologi Malaysia (03M71), and a grant from Takasago Thermal Engineering Co. Ltd., Japan (4B424).

**Institutional Review Board Statement:** Not applicable.

**Informed Consent Statement:** Not applicable.

**Data Availability Statement:** The data presented in this study are available on request from the corresponding author.

**Conflicts of Interest:** The authors declare no conflict of interest. The funders had no role in the design of the study; in the collection, analyses, or interpretation of data; in the writing of the manuscript, or in the decision to publish the results.

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
