# Peer review of "Development of a Prediction Model of the Pedestrian Mean Velocity Based on LES of Random Building Arrays"

_buildings, doi:10.3390/buildings12091362_

Round 1

Reviewer 1 Report

Comments:

Please describe the methodology  more elaborately.

Provide good quality pictures for  figures 3, 4,7.

Conclusion should be supported by some important results.

If possible provide a comparison  with the existing literatures.

The predictability  of the proposed model is still  low. Why?

Author Response

Refer attachment.

Reviewer 2 Report

This research attempts to establish the relationship between the local mean velocity ratio and the local frontal area density. The study is interesting, but Some issues need to be revised before the article is published.

1. First of all, the research method is not clear. I do not understand weather the CFD simulation was performed in this study. From figure 3, From Figure 3, it appears that the study used numerical simulations. But there is no statement about the simulation method of the large eddy simulation in the article, and its accuracy demonstration is not discussed. If the data comes from previous research, the method of obtaining it and the feasibility of applying this research need to be clearly stated. Unfortunately, I didn't find any relevant content either.

2. This article structure is very strange. In section 2, there is one subsection 2.1 Random staggered array. I don’t understand why.

3. In introduction, the author indicated that “there has not been a credible model that can predict the mean velocity ratio in an urban area exclusively around a specific targeted building”. If the study is predicting wind speed conditions around the target building, rather than the previous study discussing the overall wind conditions of the building groups. I think the study is interesting. If so, the calculation of the specific local frontal area density should be defined clearly in the section of method (e.g., the calculated area referring the targeted building).

4. The geometry setting of studied objects needs to be discussed more carefully. I don't know the basis for building height variation. Why the height of the Low-rise buildings are 0.36h and 0.84h. the height of Medium-rise and High-rise buildings varied 1.32h-2.0h and 2.64h-3.76h, respectively. What buildings do they correspond to in reality? The h is set as 0.025. How many meters is this height h in reality? In addition, the layouts of these different height buildings need more discussion.

5. In abstract, you said “Building impact on wind flows is unpredictable at pedestrian level particularly due to non-uniform building heights of a typical urban area.” This statement is not accurate enough. There are many factors of building form that can affect wind conditions. Building height variation is only on factor. Please read through the whole text and carefully consider the sentences.

6. There are some minor issues that need to be fixed. There two Table 2. some abbreviations are not required. For example, isolated roughness flow (IRF), wake interference flow (WIF) and 54 skimming flow (SF), the abbreviations are not used in the following text. Figure 6 should be described in section 2. The content of the article needs to be reorganized, especially with regard to research methods.

Round 2

Reviewer 1 Report

No further comments. The paper can be accepted for publication. 

Author Response

Thank you.

Reviewer 2 Report

Regarding the calculation of the local  frontal area density , there are still two problems that need to be clarified.

1. Building density should be local plan area density ??(t),  not the total plan area density, ??. It is especially important for urban building complexes with non-homogeneous texture forms. This leads to a question, how to determine the scope of the local area.

2. We know that the building facade area of the frontal area density density is the projected area of the windward building. How the calculation of the local frontal density in this study takes into account the blocking effect of sourround buildings.

The calculation target in Figure 2 should be clearly marked as in Figure 3.
